# The Oral Glucose Tolerance Test—Is It Time for a Change?—A Literature Review with an Emphasis on Pregnancy

**DOI:** 10.3390/jcm9113451

**Published:** 2020-10-27

**Authors:** Delia Bogdanet, Paula O’Shea, Claire Lyons, Amir Shafat, Fidelma Dunne

**Affiliations:** 1Department of Medicine, School of Medicine, National University of Ireland Galway, H91TK33 Galway, Ireland; PaulaM.OShea@hse.ie (P.O.); amir.shafat@nuigalway.ie (A.S.); fidelma.dunne@nuigalway.ie (F.D.); 2Department of Diabetes and Endocrinology, Saolta University Health Care Group (SUHCG), University Hospital Galway, H91YR71 Galway, Ireland; 3Department of Clinical Biochemistry, SUHCG, University Hospital Galway, H91YR71 Galway, Ireland; Claire.lyons11@gmail.com

**Keywords:** oral glucose tolerance test, reproducibility, diabetes, gestational diabetes

## Abstract

Globally, gestational diabetes (GDM) is increasing at an alarming rate. This increase is linked to the rise in obesity rates among women of reproductive age. GDM poses a major global health problem due to the related micro- and macro-vascular complications of subsequent Type 2 diabetes and the impact on the future health of generations through the long-term impact of GDM on both mothers and their infants. Therefore, correctly identifying subjects as having GDM is of utmost importance. The oral glucose tolerance test (OGTT) has been the mainstay for diagnosing gestational diabetes for decades. However, this test is deeply flawed. In this review, we explore a history of the OGTT, its reproducibility and the many factors that can impact its results with an emphasis on pregnancy.

## 1. Introduction

### 1.1. Diabetes and Gestational Diabetes—Historical Aspects

Diabetes mellitus has been recognised since 1500 BC [1]. Diabetes with onset during pregnancy was first described in 1824 in Germany [2]. Lambie in 1926 determined that the first manifestation of diabetes in pregnancy occurs in the 5th or 6th month of pregnancy. He advocated the use of the 50 g oral glucose challenge test (OGCT) to calculate ketogenic-anti-ketogenic balance [3]. Based on Hoet’s study [4], in 1957, Wilkerson [5] developed a protocol proposing a 3 h oral glucose tolerance test (OGTT) for patients at high risk for developing diabetes. Additionally, for women with no risk factors, he recommended a 2-step approach: 1 h blood glucose measurement after a 50 g glucose load which, if abnormal, was followed by a 3 h OGTT.

The clinical equipoise regarding the best approach to screen and diagnose gestational diabetes (GDM) was the impetus for Norbet Freinkel to organise the First International Workshop on GDM in 1979 [6]. A core outcome of this event was the emergence of a model for GDM screening and the suggestion that screening should be carried out between 24 and 28 weeks’ gestation. This model was updated in 1984 at the Second International Workshop on GDM [7], which concluded that all pregnant women should be screened for glucose intolerance with a 50 g OGCT, irrespective of the time of the last meal or time of day, and for diagnostic purposes the 100 g OGTT was to be employed. In 1990, at the Third International Workshop on GDM [8], screening and diagnostic criteria were confirmed. This panel agreed that the 75 g 2 h OGTT should be used to screen women at high risk of developing GDM [8].

The seminal Hyperglycaemia and Adverse Pregnancy Outcome (HAPO) Study in 2008 [9] addressed the importance of having all three glucose values (fasting, 1-h and 2-h post glucose load) in the OGTT since none of the glucose values were significantly correlated, and no single value was better in predicting a GDM diagnosis.

### 1.2. OGTT

The OGTT has been used in clinical medicine for over 100 years [10] and was first described by Conn [11]. His findings were based on the work of Jacobsen in 1913, who demonstrated that carbohydrate ingestion leads to glucose fluctuations [10]. Since then, the OGTT has been contested [12]. The main concerns raised by Unger in 1957 were the diagnostic values at each time point, the timing of samples, diet (at that time 300 g of carbohydrates for 3 consecutive days prior to the test was recommended), exercise, age, gastrointestinal factors (e.g., gastric emptying time or gastrointestinal absorption rates) and stress prior to the test that may influence the values of the test. In 1964, Nadon et al. [13] completed a comparative analysis between OGTT and the intravenous glucose tolerance test (IVGTT) and found considerable disagreement between both in the identification of diabetes. They concluded that, in the future, diabetes “may be diagnosed without reliance on glucose tolerance tests alone” [13].

### 1.3. Reproducibility

In 1965, McDonald et al. examined the reproducibility of the OGTT [14]. In this work, 400 male volunteers free of diabetes underwent a series of six separate OGTTs and demonstrated that blood glucose levels for individuals varied considerably. A decade later, these findings were corroborated by Olefsky et al. [15].

In 1991, Harlass et al. [16] found OGTT reproducibility of only 78% in women with an elevated glucose concentration 1 h post a glucose load when repeated within 2 weeks. Catalano [17] reported poor reproducibility for the OGTT in diagnosing GDM in 24% (9 of 38) of pregnant women tested. The authors hypothesised that this was likely due to a norepinephrine-mediated process where maternal stress leads to increased concentrations of glucose and insulin. This theory was supported by Ko et al. [18], who found the overall reproducibility of the OGTT to be 65.5% with subjects showing an improvement in glycaemic status on repeat testing. More recently, Munang et al. [19] showed the reproducibility of the OGTT for GDM in a sub-Saharan African population to be 74.2%. In this study, 70 women underwent the OGTT at 24–28 weeks of gestation and again one week later. However, the generalisability of the results of this study to other populations is questionable due to the small cohort, the short time interval between repeat testing and the fact that glucose was measured on capillary blood samples and not plasma as is more usual.

Despite scientists raising concerns about the reproducibility of the OGTT for over 50 years, it remains the only available test and the current “gold standard” for diagnosing Type 2 Diabetes Mellitus (T2DM) and GDM. In this review, the myriad of variables that affect the reproducibility and accuracy of the OGTT are discussed in terms of the Total Testing Process: pre-analytical, analytical and post-analytical phases (Figure 1).

### 1.4. Screening

Diagnosing GDM is important not only for the short-term adverse outcomes related to pregnancy and delivery, but also for the long-term consequences affecting both the mother and the child such as development of T2DM, obesity, metabolic, cardiovascular, neurological and psychiatric problems [20]. The main purpose of GDM screening is the identification of GDM cases, thus facilitating early lifestyle interventions and treatment. Randomised clinical trials (RCTs) have shown that treatment of GDM through lifestyle changes and pharmacological interventions leads to a reduction in adverse perinatal outcomes (large/small for gestational age, macrosomia, prematurity, neonatal hypoglycaemia and caesarean section delivery) [21,22].

Debate continues on the optimum screening strategy to diagnose GDM [23]. Universal screening is an approach where all pregnant women are screened. Critics of this approach highlight the fact that, if adopted, universal screening would mean that many women without GDM would be subjected to unnecessary invasive testing and that the cost implications for healthcare systems would be significant [24]. The alternative approach is selective screening based on the presence of risk factors. Selective screening is less costly as fewer women are tested. However, unscreened women may develop GDM and remain undiagnosed with the potential for increased adverse outcomes. Risk factors for GDM include age ≥30 years, family history of diabetes, increased body mass index (BMI), previous GDM, miscarriage, polycystic ovarian syndrome (PCOS) and a previous large for gestational age (LGA) or macrosomic baby [25]. The Atlantic Diabetes in Pregnancy (Atlantic DIP) study group evaluated the difference in GDM prevalence using three distinct guidelines for selective screening in a cohort of universally screened pregnant women [26]. This research found that by using 2008 National Institute for Health and Care Excellence (NICE) [27], 2010 Irish [28] and 2013 American Diabetes Association (ADA) guidelines [29], 20%, 16% and 5% of women, respectively, would have been misdiagnosed as not having GDM. In an Italian study, Pintaudi et al. [30] found that when universal screening was applied, 11.1% of pregnant women were identified as having GDM, but when selective screening was applied to the same cohort, 23% of GDM cases would have been missed.

In an effort to provide universal screening such that no case of pregnancy dysglycaemia is missed, researchers are intensifying the quest to identify an alternative biomarker/test that would easily, accurately, reproducibly and economically detect this at-risk maternal population. Identifying a minimally invasive biomarker that could be used as a single test in the non-fasting state would have clear advantages over the current fasting OGTT.

## 2. The Total Testing Process

### 2.1. Pre-Analytical Phase

The importance of the pre-analytical phase of the total testing process is often underappreciated and accounts for 46–68% of all laboratory errors [31]. An inaccurate glucose measurement due to sampling without standard timepoints can lead to a missed diagnosis of GDM or mismanagement of a patient with GDM with the potential for increased adverse outcomes and healthcare costs. This is particularly relevant when using the International Association of the Diabetes and Pregnancy Study Groups (IADPSG) criteria, as only one of three values is required to be met or exceeded to make the diagnosis.

### 2.2. Physiological Factors

#### 2.2.1. Exercise

The benefits of exercise on physical and mental health have been widely documented from improvement of cardiovascular fitness and outcomes to significant reduction in depression and anxiety [32,33]. Many researchers have looked at the impact of exercise on blood glucose levels to build evidence on the importance of exercise in the management and prevention of glucose intolerance and diabetes.

In 2007, Andersen et al. [34] showed that an exercise session carried out 14 h before having a high carbohydrate meal significantly reduced post prandial levels of glucose compared to controls (*p* ≤ 0.05). Slentz et al. [35] studied the effects of different intensities of exercise on the OGTT in individuals with prediabetes. These authors found significant reductions in fasting glucose levels only when low amount of moderate exercise and diet was combined. Higher levels of exercise were associated with improved glucose concentrations at 30 min post OGTT but was less effective when compared to the combination of diet and exercise. When overall improvement in glucose tolerance was assessed, low amounts of moderate exercise alone was determined to be half as effective as diet and exercise combined but twice as effective as high amounts of high intensity exercise. These findings are supported by Houmard et al. [36], who found that exercise sessions of low and moderate intensity have a positive effect on improving insulin sensitivity and fasting plasma glucose. These results contradict previous studies [37,38] that found no improvement in the OGTT results after moderate intensity training but noticed a 30% decrease in glucose levels on the OGTT after sustained vigorous exercise sessions.

Castleberry et al. [39] examined the impact of various workout patterns (no exercise, a single exercise session, alternative days of exercise or consecutive days of exercise) on glycaemic control on the OGTT 12–14 h post the exercise session and found that the type of exercise pattern made no difference to the glucose results.

Despite contradictory data in the literature on the length and intensity of the exercise session, physical activity influences the way our body processes nutrients. Most of the studies on this topic have been carried out on healthy subjects or individuals with diabetes and there are no studies evaluating the impact of exercise on the antepartum OGTT results. Hence, further research is required to determine whether a single exercise session prior to the antepartum OGTT lowers/improves glucose results. Such evidence is also necessary to ensure that patient preparation for the OGTT is standardised with respect to the amount of exercise, if any, pregnant women should do in the days prior to testing.

#### 2.2.2. Gastric Emptying

Absorption of glucose is negligible from the mouth and the stomach, so the ingested glucose dose cannot enter the blood compartment until it is emptied from the stomach, digested to monosaccharides and transported across the intestinal epithelia. The ability of the small and large bowel for transport far exceeds the rate of delivery of the 75 g glucose challenge, and so a major rate limiting step for the absorption of glucose is gastric emptying rate. Gastric emptying in one of the main factors influencing the glucose response in the first hour after the OGTT or after a meal and is responsible for 30–35% of the variability in post-prandial glycaemia in healthy controls [40,41] and diabetic patients [42]. This supports the hypothesis that an augmentation in the volume or reduction in the osmolality of a meal may result in an intensification in the speed of gastric emptying with a consequent rise in glucose [43]. Studies have shown that the faster the gastric emptying post glucose load, the higher the postprandial glucose levels will be [40,44]. Horowitz et al. [40] found that the 2 h glucose level post OGTT was inversely related to the gastric emptying rate—the slower the gastric emptying, the higher the 2 h blood glucose level. Their hypothesis for this finding was that high blood glucose levels may influence gastric motility, slowing it down in order to reduce further glucose absorption.

We cannot control (but should always consider) the individual variability of the rate of gastric emptying. Guidelines recommend the glucose load should be drank slowly over a period of 5 min. However, this is difficult to achieve and control for in clinical practice with individual wide variations in the glucose load drinking time.

#### 2.2.3. Hydration

Research into the impact of hydration status on glycemic levels is limited. In 2015, Murry [45] explored the effects of mild hypohydration on glucose tolerance within individuals diagnosed with T2DM by evaluating blood glucose levels over two 120-min time periods in euhydrated and hypo-hydrated states, respectively. He found that reduced water consumption resulted in increased glucose concentration before and during the OGTT. Johnson et al. [46] found similar findings, concluding that 3 days of decreased total water intake in people with T2DM acutely modifies blood glucose levels during an OGTT with higher glucose measurements in the hypohydrated group. In 2016, Caroll et al. [47] piloted a study (*n* = 5) in which ~12 h hypo-hydration (sauna plus fluid restriction) induced a higher glycaemic response to a glucose load compared with sauna plus rehydration. The same group [48] however, 3 years later, found contradictory evidence indicating that acute hypohydration did not modify the glycaemic response, suggesting that when OGTTs are done in healthy subjects, hydration status may not necessarily influence the glycemic response during the OGTT. In 2019, Jansen et al. [49] conducted a crossover trial looking at the acute effect of osmotically stimulated arginine vasopressin (AVP) on glucose response in 60 healthy adults and found that acute osmotic stimulation increased glucose levels during the OGTT.

Additional findings that might reflect the importance of hydration status come from studies assessing the impact of seasonal variation on the OGTT and GDM prevalence. Numerous studies [50,51,52,53] have had consistent findings of higher GDM prevalence during the summer months with higher 1 h and 2 h values on the OGTT and no impact on fasting glucose levels. While these results can be attributed to other seasonal factors such as nutrition quality, exercise level, light-sensitive hormones or increased blood flow due to increased temperature, hydration status may be more likely to explain these higher glucose values observed. The rationale being that where increased temperature is not accompanied by adequate fluid intake, this could lead to hypohydration, hypovolemia and increased glucose concentration.

Even though there are no studies examining the impact of hydration status on the OGTT in pregnancy, we can extrapolate on previous findings and consider that the effects of hypo-hydration or hyper-hydration are not negligible and have the potential to lead to a misdiagnosis. Currently, there are no guidelines on water intake in the days prior to the OGTT.

#### 2.2.4. Stress and Sleep

In 1991, Spirito et al. [54] explored the impact of stress and coping mechanisms on glucose levels in 72 pregnant women without diabetes (mean age 27.8 years) and 125 women (mean age 27.7 years) with GDM. While levels of emotional distress and methods of coping did not show any significant difference between groups, disconnection and detachment significantly influenced daily blood glucose variability. In 2011, Hosler et al. [55] found that having any number of stressors one year prior to delivery was significantly associated with pregnant women failing their glucose test. One of the possible explanations for this is that psychological stress alters the hypothalamic-pituitary adrenocortical system and stimulates the release of cortisol thus increasing glucose levels. Importantly, both GDM studies were retrospective cohort studies, therefore the knowledge of one’s GDM diagnosis could potentially influence the woman’s disease awareness creating a recall bias for stressors.

Experiencing acute psychological stress is associated with hyperglycemia and increased risk of T2DM and glucose intolerance [56]. In 2016, Horsch et al. [57] found that intense stress (major life events) and psychological stress responses (depression, anxiety and sleep length) led to increased glucose levels during pregnancy even prior to women being tested for GDM. The variables that were associated most with increased levels of fasting blood glucose were increased distress and short sleep duration. The association between sleep duration and quality and glucose homeostasis has been highlighted by additional studies [58,59] that found that shorter sleep duration is associated with higher glucose levels, particularly the fasting and 2 h glucose level on the OGTT. Retrukatul et al. [60] found that pregnant women with reduced sleep duration (less than 7 h per night) have an increased risk of developing GDM; in fact, each hour of reduced sleep leads to a 4% increase in blood glucose levels. These results are supported by Myoga et al. [61], who also found that pregnant women who sleep less than 5 h per night had higher random blood glucose levels.

Stress can impact glycaemic status not only through hormonal responses but also through the development of unhealthy lifestyle behaviours such as overeating, smoking, increased alcohol intake [62,63]. Given the glucose response to stress and to decreased sleep duration/quality, it would seem possible that pregnant women could be erroneously diagnosed as having GDM using the OGTT.

### 2.3. Pre-Testing Patient Preparation Factors

#### 2.3.1. Length of Time Spent in the Fasting State

A regular meal can significantly influence glucose levels [64]. Similarly, fasting also influences the levels of glucose. Salehi et al. [65] noticed a significant decrease in glucose after a complete period of fasting during Ramadan of 13 h in young healthy males, while Saada et al. [66] found that glucose levels increased significantly after 10–12 h fast in women with T2DM.

The OGTT is performed after an overnight fast. However, the period of fasting is not standardised and varies between 8 h and 16 h. Despite a low level of evidence (grade B), the ADA guidelines recommend that the glucose sample should be taken in the morning, after a period of fasting of at least 8 h, with no constraints on the amount of water allowed to be consumed by the patient during this time [67,68].

Variation in the period of fasting prior to testing may influence OGTT results. In 2011, Moebus et al. [69] challenged the necessity for fasting >8 h and found that a fasting length of 3 h was adequate for a reliable glucose measurement. In the British Regional Heart Study [70], a cross-sectional study of men aged between 60–79 years, there was no difference in plasma glucose levels in those fasting for 6 h or ≥6 h.

Patient adherence to instructions for fasting prior to the OGTT must also be considered. In 2013, Kackov et al. [71] explored how well patients were informed regarding the fasting protocol for laboratory blood testing and whether patients arrived for phlebotomy appropriately prepared for testing. These authors found that 46% of the participants believed that the precise time of their last meal prior to fasting was unimportant, as long as the last meal was on the day prior to the blood test. Notwithstanding, only 60% of participants arrived for blood testing having adhered to instructions for fasting. Furthermore, 52% of study participants had not been informed about the pre-testing preparation requirements for blood testing.

Therefore, while there is no clear evidence regarding impact of the duration of time spent fasting prior to the OGTT, it is critical to standardise the duration spent fasting prior to laboratory OGTT and to give clear, consistent instructions to our patients to prevent inaccurate results.

#### 2.3.2. Preparatory Diet

Many centers recommend that the OGTT is preceded by a 3-day diet of 150 g carbohydrate per day. The concept of this is based on the original work of Conn [11]. The length of the diet preceding the OGTT and the quantity of carbohydrate recommended have been randomly selected. Conn’s 3-day diet contained in fact 300 g carbohydrate per day. Conn showed that by keeping a low-carbohydrate diet prior to the glucose test the number of false-positive cases of diabetes would increase. However, this study was small and only 3 of the 9 study participants were women.

The Fourth International Workshop-Conference on GDM [72] recommended the 3-day diet with a minimum of 150 g carbohydrate per day prior to the OGTT in order to prevent patients being misdiagnosed as having diabetes.

However, other studies [73,74,75] found that the carbohydrate ratio of the diet prior to the OGTT did not impact upon the test results indicating that a specific diet prior to the OGTT is not mandatory for women with normal dietary habits.

There is not enough evidence to recommend a pre-set diet/carbohydrate intake prior to the OGTT. Perhaps maintaining one’s normal, regular diet prior to undergoing the OGTT would best reflect the individual’s capacity to metabolise glucose. However, in order to maintain a standardised approach to OGTT, adherence to current guidelines should be recommend for now.

#### 2.3.3. Glucose Load

In 1998, Sievenpiper et al. [76] investigated the post-prandial glycaemic response (PGR) after the ingestion of 25 g glucose, sucrose or fructose dissolved in either 200 mL or 600 mL of water. They established that PGR was not only influenced by carbohydrate type but also by the volume dose. By increasing the meal volume from 200 mL to 600 mL, PGR areas were significantly increased for all three sugars. Building on these results, Sievenpiper investigated the effects of a 2- and 3-fold increase in the volume of a 300 mL 75-g OGTT on glycaemic concentrations [77]. He found that there was a significant statistical difference between the means of the area under the curve (AUC) for the 300 mL, 600 mL and 900 mL OGTTs (*p* = 0.006). While post prandial glucose levels were not affected by the increase in volume from 300 mL to 600 mL, glucose levels were significantly increased when the volume was increased to 900 mL.

Fifty years ago, in an effort to reframe and strengthen this analysis, the Committee of the Statistics of the American Diabetes Association (ADA) suggested that the glucose load used during the OGTT should be based on an estimation of the individual body surface area (BSA) [78]; However, despite this recommendation, in 1980 the World Health Organization (WHO) endorsed the use of the 75 g glucose load for the OGTT irrespective of the individual BSA [79,80]. Subsequently, the glucose dose was set at 1.75 g/kg body weight with a maximum of 75 g [72,81]. Practically, this means that all patients over 43 kg are tested using the maximal dose of 75 g glucose. Furthermore, a number of studies have shown an association between a person’s height and their 2 h glucose values on the OGTT [82,83], which supports the ADA’s findings. In 2019, Palmu et al. [84] showed that the BSA has a considerable impact on the blood glucose levels from a standardised 75 g OGTT, with smaller individuals more likely to be diagnosed with diabetes or glucose intolerance compared to individuals with a larger BSA.

Therefore, emerging research has made a strong case for glucose loading to be individualised according to BSA. Additionally, research is steering practitioners to reexamine if 75 g glucose load is an appropriate dose regardless of the patient’s physical characteristics. Indicators are showing that the glucose values following a 75 g glucose load is expected to differ according to variable factors such as pancreatic beta cell function, gut hormones and neural responses to carbohydrate ingestion. The problem becomes clear when individuals with a small BSA are diagnosed with diabetes or glucose intolerance, despite their daily glucose values not exceeding the diabetes threshold. Moreover, individuals with an increased BSA might not reveal an abnormal glucose response, even though their daily glucose values meet the diabetes diagnostic criteria because the 75 g glucose dose is inadequate to increase the glucose level to ≥11.1 mmol/L compared to their normal daily caloric intake required to maintain their BMI. Consequently, the parameters for loading dose of glucose in the tolerance test should ideally be individualised according to BSA, activity level, or necessary caloric intake calculated for the individuals basal metabolic rate in order to increase its usefulness in the identification of glucose intolerance.

There are several options regarding the preparation and delivery of the standard 75 g glucose load. One of the most popular options has been the use of Lucozade (Energy Original), which contained 70 kcal and 17 g of carbohydrates per 100 mL. To obtain 75 g of glucose required the consumption of 410 mL. The current reformulated product has a ~50% reduction in calories (available April 2017), contains 37 kcal and 8.7 g of carbohydrate per 100 mL, and to deliver a 75 g glucose load requires the consumption of a large volume, 860 mL. This change in formulation of Lucozade makes it unsuitable for use in the OGTT. To overcome this issue, an alternative form of 75 g anhydrous glucose (glucose monohydrate 82.5 g) comes in powder form in a ready-to-use sachet. It requires dissolving in 250 mL of water (to give final volume of 300 mL). Polycal^®^ (Nutricia) (Nutricia Ltd., White Horse Business Park, Newmarket Avenue, Trowbridge, Wiltshire, UK)comes in liquid form and necessitates having access to a sufficiently accurate measuring vessel to accurately measure out 113 mL (equivalent to 75 g glucose) to which water is then added and mixed to give a final total volume of 250–300 mL. Rapilose^®^ OGTT Solution (Penlan Healthcare Ltd., Abbey House, Wellington Way, Weybridge, UK) comes in liquid form and is available in a ready-to-use 300 mL pouch containing 75 g anhydrous glucose. Rapilose^®^ has be customised for patients with a body weight ≥43 kg where they should consume the entire contents of one pouch but patients who weigh under 43 kg should have the volume adjusted accordingly.

### 2.4. Pre-Analysis Sample Handling

#### 2.4.1. Sampling Site

In order to improve the interpretation of glucose results, it is imperative to understand the difference in results between samples collected from different sites (capillary plasma, capillary whole blood, venous plasma and venous whole blood). For example, the glucose levels in plasma are 11% higher than the levels in whole blood despite the fact that in clinical practice the words “plasma” and “blood” are used interchangeably [85].

Under normal physiological conditions, the post-prandial, capillary glucose levels are higher than the venous glucose levels as determined by the rate at which glucose is extracted from blood by tissues. Exploring this anomaly, Kuwa et al. [86] examined the difference in glucose levels between capillary and venous samples during the OGTT in 75 healthy individuals. They found that venous and capillary glucose levels were comparable in the fasting state, but the post-load capillary sample had significantly higher glucose levels compared to the venous one.

Stahl et al. [87] investigated whether capillary whole blood glucose levels used for analysis can be expressed as plasma results (as recommended by the ADA and WHO). Results from this study confirm that translation from capillary to plasma values may be acceptable for mean values but should not be used for individual glucose levels. These findings were confirmed by Colagiuri et al. [88], assessing the correlation between glucose levels in capillary and venous samples in fasting state, 2 h after oral glucose load and random glucose levels. These authors established that both fasting and random capillary samples gave lower glucose values than venous samples but the 2 h post glucose load capillary sample gave higher glucose values than the venous sample.

Adding to these research conclusions, D’Orazio et al. [89] maintain that due to the difference in glucose concentrations observed between whole blood and plasma, the glucose levels are not interchangeable. They also recommend that the reporting of glucose measurements should be in plasma only as the concentration of glucose in plasma is independent of hematocrit.

Preferably, the best model is one where blood glucose levels are reported from plasma samples where glycolysis has been delayed or inhibited. Alternatively, glucose level measurement reports should have clear information on the sample type being used and if any conversion factors had been applied in the reporting process.

#### 2.4.2. Specimen Collection Tube

A prominent source of pre-analytical error in determining plasma glucose levels in vitro is glycolysis. It is reported that glycolysis leads to 5–7% decrease in glucose levels per hour at room temperature [68]. There are two main approaches to inhibit glycolysis. The first requires immediate separation of plasma/serum (within 30 min of sampling) from blood cells prior to analysis. The second approach involves collecting venous whole blood into specimen tubes containing a glycolytic inhibitor.

Sodium fluoride is one such glycolytic inhibitor and acts to inhibit enolase activity [90] stabilising the glucose levels in the long term. However, enolase is late in the glycolytic pathway such that glycolysis continues during the first hours after the sample has been collected. The rate of glucose loss is similar during the first 90 min regardless of the presence of sodium fluoride [68,91]. Furthermore, using sodium fluoride as a glycolytic inhibitor leads to an error in glucose levels that ranges between 0.28 and 0.39 mmol/L (5–7 mg/dL), and can be as high as 1.1 mmol/L (20 mg/dL) if plasma is left unseparated for more than 3 h post collection [92]. These findings are supported by Chen et al. [93] who confirmed the failure of sodium fluoride to inhibit glycolysis one hour after sample collection and recommending that the best way to reduce glycolysis and improve glucose integrity in samples in vitro was through immediate separation of plasma from blood cells.

Therefore, using sodium fluoride alone as a glycolytic inhibitor is considered insufficient. To circumvent this issue, Uchida et al. [94] showed that acidification quickly inhibits glycolysis through the inhibition of hexokinase and phosphofructokinase. In 2013, Garcia del Pino et al. [95] determined that citric acid immediately inhibits glycolysis. These authors showed that the glucose levels in samples taken in sodium fluoride tubes was significantly lower when compared to the glucose levels taken in temporally paired citrate tubes. Comparable results were reported by Norman et al. [96] evaluating paired fasting plasma glucose samples collected into sodium fluoride and citrate tubes and found higher glucose levels in the samples collected into the citrate tubes. This was reaffirmed in 2019 by Jamieson et al. [97], seeking to compare plasma glucose stability over time in 501 samples taken at the time of the OGTT after 24 weeks of gestation and found that the samples containing citrate as a glycolytic inhibitor offered the best short and long-term stability for glucose levels even compared with fluoride samples placed immediately on ice. They suggested that the use of sample tubes containing citrate would not require services to make any changes in the sample collection protocols (such as the addition of ice or immediate plasma separation). However, the authors advised that the diagnostic criteria for glucose intolerance may need revision as glucose values were, on average, 0.2 mmol/L higher when using fluoride-citrate sample tubes compared to those obtained by research methodology. In support of these findings, Lyons et al., 2018, assessed the stability of glucose in citrate-fluoride-oxalate buffered plasma (FC-Mix tubes) stored at 4 °C and 18–22 °C for 8.5 days and found that glucose results were maintained within 0.20 mmol/L of those determined using WHO specifications [98].

In clinical practice, where delays of sample transport and processing are regularly encountered, the use of citrate tubes delivers the best option in inhibiting glycolysis and preserving the integrity of blood glucose levels ex vivo. The use of citrate buffered specimen tubes is recommended by the ADA especially if the sample processing is likely to be more than 30 min post-collection [68].

#### 2.4.3. Sample Storage and Transport

In 1985, the WHO recommended “rapid plasma separation from samples collected in fluoride tubes” in order to prevent or delay glycolysis [99]. The AACC and ADA guideline [68] recommends that samples “be immediately immersed in an ice-slurry and analyzed within 30 min of collection or rapid centrifugation after collection”. However, compliance with these guidelines is particularly challenging in the case of the OGTT due to that fact that fasting and post glucose samples are usually held at the point of patient care until the test is completed, invariably over 2 h.

Consequently, diabetes prevalence will be underestimated in research studies in which sample handling and analysis is delayed as indicated by Potter et al. [100], who compared OGTT results (sodium fluoride tubes) between early centrifugation (within 10 min) and delayed centrifugation (at the end of the OGTT test) in over 12,000 women. They found the mean glucose levels for fasting, 1 h, and 2 h OGTT samples were higher using early centrifugation (*p* < 0.0001 for all) compared to delayed processing, increasing the GDM prevalence from 11.6% (*n* = 869/7509) to 20.6% (*n* = 1007/4887). In the commentary accompanying this study, Price et al. [101] highlight that “without strict pre-analytical OGTT sample handling in routine clinical practice, our ability to accurately diagnose GDM and report GDM prevalence data will be flawed”.

The pre-analytical blood sampling protocol for pregnancy OGTT requires revision and standardisation [102]. Consideration of the difficulties that rapid centrifugation (within 30 min of sampling) or placement of samples on ice in busy clinics illustrates that value and pragmatism of the use of citrate blood tubes for sample collection. However, the use of citrate tubes has the potential to give a positive bias of 0.2 mmol/L, falsely increasing the rate of GDM diagnosis, such that a correction factor or revision of the diagnostic thresholds may be required [96,103,104]. An alternative approach could be to measure glucose in lithium heparin plasma analysed on the critical care analyser at the point of care (POC). In 2018, Lyons et al., recruited 12 volunteers to undergo the OGTT measuring blood glucose at each time point on the critical care analyser (ABL90FLEX^®^/Glucose oxidase), (Manor Court, Manor Royal, Crawley, West Sussex, England) and concomitantly in whole blood collected into fluoride-oxalate tubes immersed immediately in ice-slurry and analysed within 30 min using the central laboratory (Roche Cobas^®^ 8000 modular analyzer series/Hexokinase) (Roche Diagnostics GmbH, Sandhoferstrasse 116, Mannheim, Germany) [105]. These authors demonstrated good agreement of glucose results with the WHO recommended method with results within the total allowable error analytical goal for plasma glucose of < 5.5%.

While clear recommendations exist regarding glucose sample transport and storage, the challenge is the practicality and applicability of these guidelines to the routine clinical practice settings that are not resourced for immediate sample handling and processing. Outside of research specific laboratories, worldwide, very few centers are likely equipped to adhere to such strict glucose processing methodology. Citrate buffered specimen tubes offer the best practical solution and their use is recommended by the ADA.

### 2.5. Analytical Phase

#### 2.5.1. Traceability and Methodology

##### Central Laboratory

Global standardisation of clinical assay’s aims to produce accurate and reproducible test results across space and time (traceability) through a reduction in method variability. To minimise assay bias, methods for measuring glucose should be calibrated (traceable) to reference methods. Currently, there are two reference methods for blood glucose measurement recommended by the Joint Committee for Traceability in Laboratory Medicine: isotope dilution mass spectrometry (ID-MS) [106], and enzymatic (Hexokinase-Glucose-6-Phosphate Dehydrogenase) [107]. The maximum allowable deviation for the alignment of the central laboratory method with a reference method is 4%. In the routine clinical central laboratory setting, glucose is invariably measured using one of three common enzymatic methods: hexokinase, glucose 1-dehydrogenase and glucose oxidase in reactions that either are coupled to a chromophore or generate an electric current.

#### 2.5.2. Point of Care (POC)

##### Blood Glucose Meters (BGM)

Glucose is measured using capillary blood glucose concentrations. All POC meters use enzymes to measure glucose. These enzymes are oxidoreductases, can be classified in several categories each with its specific characteristics but, ultimately, have as a primary role to oxidise glucose [108]. Electron transfer to an electrode is then measured (third generation sensors). Of note, none are completely specific for glucose.

In 2020, O’Malley and colleagues [109] studied the use of POC glucose measurements in diagnosing GDM in women undergoing an antepartum OGTT. These authors found the diagnostic accuracy of POC glucose for GDM to be 83.0% (95% confidence interval (CI), 74.2–89.8) and concluded that there is no justification for the use of POC in centers that have adequate sample handling facilities. However, they noted that POC might be acceptable in low- and medium-resource settings, where processes to inhibit glycolysis are not available.

##### Critical Care Analysers (Blood Gas Analysers)

Whole blood (venous/arterial) is collected into balanced sodium heparinised (plasma) syringes and glucose is measured by a fixed enzyme electrode or via a reagent cassette. Glucose oxidase is the enzyme most commonly used [110].

### 2.6. Analytical Quality

#### 2.6.1. Central Laboratory

Generally, laboratory testing quality should not be one of the variables influencing GDM prevalence and should not cause any glucose variability though the measurement process. The total laboratory analytical error has two main components: (1) precision, which is the capacity of the test to reproduce replicate measurements and it is expressed as the coefficient of variation (CV); and (2) bias, which is the difference between the laboratory result and the true value of the test. A good laboratory test should have minimal imprecision and bias and should conform with the specified analytical regulatory criteria. Laboratories compare their test result and the performance of their measurements against objective quality requirements such as the National Academy of Clinical Biochemistry (NACB) guidelines for total maximum allowable error (TEa). For glucose, the recommended targets are imprecision < 2.9%, bias < 2.2% and TEa < 6.9% [68]. The analytical imprecision for central laboratories is of the order of 1–2%.

However, glucose measurements, even within permissible limits, can influence GDM incidence and prevalence significantly. The true value of a laboratory test ranges within a 95% confidence interval of the reported value. In 2015, Agarwal et al. [111] examined the impact the analytical quality of a laboratory can have on GDM prevalence by comparing the total analytical error in one laboratory with the TEa as recommended by the NACB. This was a prospective study with over 2000 study participants. The research team found that, irrespective of criteria used to diagnose GDM (IADPSG, ADA, CDA), the analytical variation in glucose measurement had both a statistically significant impact on the GDM prevalence and also a significant impact on pregnant women that would be incorrectly reassured as not having GDM. These authors suggest that laboratories with decreased quality performance that report glucose measurements outside the 95% CI will ultimately lead to an increased reported GDM prevalence and an increase in false positive GDM cases. Based on total analytical variation of glucose for glucose in the laboratory performing the analyses, in their cohort, the prevalence of GDM ranged from 27% to 71% with an absolute prevalence of 45.3% (independent of the diagnostic criteria used). The authors concluded that the reported GDM prevalence has the potential to vary from 0.5–2.0-fold even if the laboratory meets the NACB recommendation of TEa < 6.9%, and urge laboratories to strive to improve their analytical performance even beyond the NACB recommendation in order to avoid misclassifying patients. Supporting these findings, Nielsen et al. [112] found that at 0% bias, an increase in imprecision from 2.7% to 3.7% increased the prevalence of diabetes by 90%.

Clinicians should always seek to use accredited laboratories but must be aware that TEa does not take into account pre-analytical factors that may influence glucose results. For example, a delay of more than 4 h in processing (centrifugation and separation of plasma from blood cells) the fasting sample of the OGTT would exceed the TEa for glucose.

#### 2.6.2. POC—BGM

The analytical variation for BGM is commonly of the order of 5%. POC guidelines recommend that 95% of glucose results from BGM should be within ± 12.5% of the central laboratory glucose results ≥ 5.55 mmol/L (100 mg/dL) and within 0.67 mmol/L (12 mg/dL) for values < 5.55 mmol/L (100 mg/dL); furthermore, that 98% of BGM glucose results should be within ± 20% of the central laboratory glucose values ≥ 4.2 mmol/L (75 mg/dL) and within ± 0.83 mmol/L (15 mg/dL) for glucose values < 4.2 mmol/L (75 mg/dL) [113].

#### 2.6.3. POC—Critical Care Analysers (Blood Gas Analysers)

The analytical imprecision for critical care analysers is of the order of 1–2% and similar to that of the central laboratory [114,115].

### 2.7. Post-Analytical Phase

The next phase of the total testing process is the post-analytical phase, which includes the following steps:

Processing of results into a report format (paper or electronic).

Identification of critical results and communication to the requesting clinician.

Interpretation of the results and if deemed necessary provision of advice for further tests.

Transmission of final report to the requesting clinician.

In the context of the OGTT, the diagnostic criteria are not uniform and are the subject of much debate.

In 1964, O’Sullivan et al. [116] proposed that “screening, diagnosis and treatment of hyperglycaemia in women who are not known to have diabetes improves outcomes”. The diagnostic criteria proposed were based on the 3 h–100 g glucose OGTT, which were subsequently validated for the development of future maternal T2DM [116]. The values proposed for GDM diagnosis were: fasting, 6.1 mmol/L (110 mg/dL); 1 h, 9.4 mmol/L (170 mg/dL); 2 h, 6.7 mmol/L (120 mg/dL) and 3 h, 6.1 mmol/L (110 mg/dL). Women with at least two abnormal values were diagnosed with GDM.

In 2008, the HAPO study showed that mild hyperglycemia was associated with adverse neonatal outcomes even below the previous GDM diagnostic criteria [9]. Based on these findings, in 2010 the IADPSG recommended a one-step 75 g OGTT and modified the GDM diagnostic cut-off points: fasting glucose: 5.1 mmol/L, 1 h glucose: 10.0 mmol/L and 2 h glucose: 8.5 mmol/L (fasting glucose: 92 mg/dL, 1 h glucose: 180 mg/dL and 2 h glucose: 153 mg/L) [81]. A single abnormal value confirms a diagnosis of GDM. Some critics of the new IADPSG diagnostic criteria indicate that the HAPO study did not take into account all pre-specified adverse outcomes and factors such as the rates of cesarean section or neonatal hypoglycemia in the determination of diagnostic cut-off points. Another criticism was that the single abnormal value required for diagnosis and the low glucose threshold of the new criteria to identify women as having GDM meant that such women would be in a very low risk category [117].

The IADPSG criteria were embraced by many international organisations including the ADA [118], WHO [119], the International Federation of Gynaecology and Obstetrics (FIGO) [120] and European Board and College of Obstetrics and Gynaecology (EBCOG) (2015). At the same time, some international bodies have not incorporated the IADPSG criteria: American College of Obstetricians and Gynecologists (ACOG) Practice Bulletin [121], the National Institutes of Health (NIH) consensus statement [122] and Society of Obstetricians and Gynaecologists of Canada (SOGC) [123]. The reasons given for not adopting the IADPSG criteria were (1) the benefit of treating women with mild GDM is not well established; (2) the increased prevalence of GDM will lead to additional healthcare costs; (3) caesarean section delivery and neonatal intensive care unit (NICU) admission rates will increase; and (4) patients identified as having GDM will develop additional psychosocial burdens which will decrease their Quality of Life (QoL).

Inconsistencies in GDM diagnostic criteria worldwide have led to challenges in making meaningful comparisons between study results (through systematic reviews and metanalysis). Cost analysis studies should always include clinical adverse outcome prevention through diagnosis and treatment in their analysis. A very well designed study by Duran et al. [124] found that the use of the IADPSG criteria was associated with an improvement in the prevalence of maternal and neonatal adverse outcomes (pregnancy induced hypertension, prematurity, caesarean sections, NICU admissions, LGA and SGA) that was cost-effective despite a 3.5-fold rise in GDM prevalence.

#### 2.7.1. COVID-19: Implications for GDM Testing

In the context of the coronavirus 2019 (COVID-19) pandemic, travel restrictions, the time (up to 3 h) spent in a potentially infectious environment while the OGTT is carried out and the requisite glucose samples collected, together with the additional number of clinical visits consequent to a positive GDM diagnosis, have combined to reduce the use of the OGTT. In fact, McIntyre et al. [125], have highlighted that international bodies have already moved to using one or more of the following alternative approaches to GDM diagnosis: fasting venous plasma glucose [126], random venous plasma glucose and/or HbA_1c_ [127]. Unfortunately, both approaches, while safer in the context of the SARS-CoV-2 pandemic, will lead to many women with GDM not being diagnosed. Gemert et al. [128] have shown that by only using a fasting plasma glucose ≤ 4.6 mmol/L for the diagnosis of GDM, 29% of women would have been missed. Similarly, van-de- l’Isle et al. [129] found that by using the Royal College of Obstetrics and Gynecologists recommendations for the diagnosis of GDM (fasting glucose ≥ 5.3 mmol/L or HbA_1c_ ≥ 39 mmol/mol or random plasma glucose ≥ 9 mmol/L), 57% of women would have been wrongly diagnosed as not having GDM. The likely consequence of this will be an increase in GDM-related complications as these women will not have received the appropriate treatment for GDM. In their commentary, Mcintyre et al. [130] emphasise the need for validation and regulatory approval of alternative, less cumbersome strategies for the diagnosis and classification of GDM by using new non-fasting biomarkers such as plasma glycated CD59, a complement regulatory protein, which is showing promise. The need for change to the way in which the diagnosis of GDM is made has been recognised for many decades now. The current global COVID-19 pandemic has reignited the urgent quest for the rapid identification of a new, reliable and feasible biomarker to diagnose GDM.

#### 2.7.2. Emerging Biomarkers

The current COVID-19 pandemic has highlighted what the scientific community has known for years [13]—that it is time to identify new tests that can accurately and robustly diagnose GDM, tests that require less preparatory and sampling time and that are less affected, if at all, by the pre-analytical factors mentioned in this article. There are now several biomarkers showing great potential to meet this clinical need. They include amino acids, peptides, proteins, lipids, enzymes, saccharides, microRNA, etc. The following biomarkers are a cross-section of the emerging data in this field.

Adiponectin is a protein hormone and adipokine involved in glucose metabolism. Many researchers have shown that adiponectin levels can diagnose GDM and can also predict GDM when analysed in early pregnancy. In 2008, Lain et al. [131] showed that women with a low first trimester level of adiponectin were 10 times more likely to be diagnosed with GDM later in pregnancy. In 2013, Rasanen et al. [132] supported this work showing that first trimester adiponectin levels were associated with the development of GDM. Additionally, an Irish study [133] found that high first trimester adiponectin levels were associated with a reduced risk of developing GDM validating the work of Rasanen et al. Furthermore, there is evidence to suggest that adiponectin may also be used in predicting the development of post-partum glucose intolerance in women with a history of GDM [134]. However, despite these promising results, in 2016 a study by Iliodromiti et al. [135] found the sensitivity and specificity of adiponectin in predicting GDM diagnosis to be 60.3% and 81.3%, respectively. Prospective studies to confirm the adiponectin role as a GDM diagnostic biomarker are warranted.

Emerging research on GDM has shown that CD59, a glycoprotein biomarker, has the potential to diagnose GDM. Gosh et al. [136] determined that glycated CD59 (gCD59) accurately predicted the development of GDM with a sensitivity of 85% and a specificity of 92%. These authors found that GDM patients had 10-fold higher levels of gCD59 compared to controls. These findings are supported in work by Ma et al. [137], showing that gCD59 levels in pregnant women before 20 weeks of pregnancy accurately predict the results of the OGTT. In addition, gCD59 levels were also associated with higher risk of delivering a baby large for gestational age (LGA). Prospective studies are ongoing to assess the potential of gCD59 to identify GDM early in pregnancy and improve prediction of adverse pregnancy outcomes [138].

Extracellular vesicles (EV) have also shown diagnostic potential as indicated in a study by Salomon et al. [139], who found a 2-fold higher concentration of exosomes (small EV) in GDM pregnancies compared to normal pregnancies. These findings have been further supported by several recent studies [140,141] which found higher concentrations of EV in women who developed GDM compared to controls.(add references) Jayabala et al. [142] examined the differences in protein content in EVs between women with GDM and women with normal glucose tolerance. They found a total of 78 proteins that were significantly differentially expressed in GDM women compared to women with normal glucose tolerance. Despite this, there are no studies comparing the levels of EV concentrations between different types of pregnancy complications (gestational hypertension/preeclampsia, foetal growth abnormalities, foetal malformations, etc). There are no studies assessing the trimester-specific EV levels in normal and GDM pregnancies or studies assessing the robustness of the test by comparing different analysis method, different methods for purifying and separating exosomes or different types of blood sample used, nor also studies investigating concentrations of EV released from placenta vs. non-placental sources. EV are an emerging biomarker with great potential; however, further studies are required to establish the exact role of EV in GDM diagnosis.

Nesfatin-1 is a polypeptide involved in food regulation and water intake and has a glucose-dependent insulinotropic action. In 2012, Aslan et al. [143] found lower nesfatin-1 levels in GDM women compared to controls. These findings are supported by Kucukler et al. [144], who similarly found lower levels of nesfatin-1 in women who developed GDM compared to women without GDM but also found a positive correlation between nesfatin-1 and insulin levels. In a recent prospective study, Mierzynski et al. [145] also observed that women with GDM had significantly lower levels of nesfatin-1 compared to women with normal glucose tolerance but also found a strong correlation between nesfatin-1 levels and pre-pregnancy BMI. However, a study by Zang et al. [146] found opposite results with nesfatin-1 levels higher in GDM patients compared to controls with a positive correlation between nesfatin-1 levels and BMI, while Deniz et al. [147] showed a negative correlation between nesfatin-1 levels and BMI. The discrepancy of these results might arise from the different study participants’ characteristics or the timing of the sample collection. While it is clear that nesfatin-1 plays a role in GDM pathophysiology, further studies are required to define its potential as a diagnostic biomarker.

Several biomarkers show potential advantages over the historical OGTT. The breadth of this emerging trend shows the activity within the research community to identify an appropriate new test for GDM diagnosis. Those mentioned above provide a mere snapshot of the evolving evidence, and an in-depth analysis is beyond the scope of this paper. This review has explored the fallacy of the OGTT, the current “gold standard” for GDM, a test that is easily affected by many variables, potentially leading to false results and has drawn attention to promising emerging alternative biomarkers. Through ongoing collaboration, researchers in this field can make a critical breakthrough on a new test for GDM diagnosis.

## 3. Conclusions

The OGTT is subject to several factors spanning the total testing process that have the potential to influence its results and negatively impact patient care. Clear guidance is needed to ensure a universal standardised approach to performing and interpreting the OGTT for the diagnosis of GDM. This will permit global harmonisation of the detection of GDM, improve the accuracy and reproducibility of the OGTT and provide for better outcomes for mothers and their offspring. Alongside this, the search for better biomarkers to diagnose GDM and ultimately replace the OGTT is gaining pace with several biomarkers currently under evaluation. However, the diagnostic accuracy and clinical usefulness of many of these novel biomarkers remain to be fully validated.

## Figures and Tables

**Figure 1 jcm-09-03451-f001:**
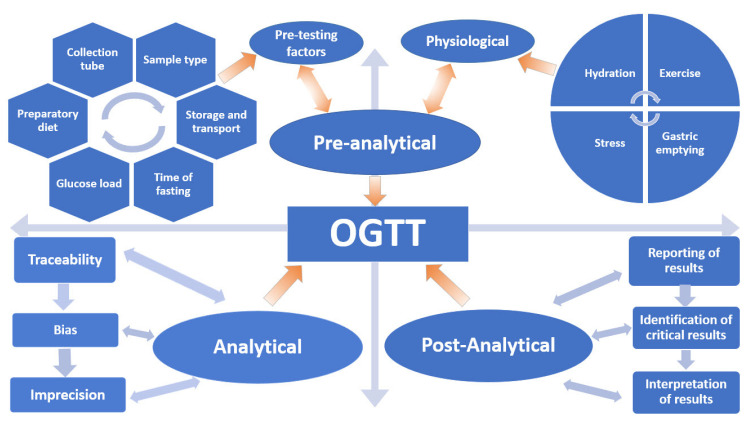
Variables that influence the oral glucose tolerance test (OGTT).

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
