# Peer review of "The Oral Glucose Tolerance Test—Is It Time for a Change?—A Literature Review with an Emphasis on Pregnancy"

_jcm, 2020, doi:10.3390/jcm9113451_

Round 1

Reviewer 1 Report

The current manuscript is well-researched, well-written and timely. The question is what is really want to tell the reader in current times. What's the the use of this paper?

Figure 1 is clear and basically the basis and nucleus of this paper and could be a guide to the practising clinician (or lab official) to critically assess their own way of working. That is a great benefit of this paper.

However, the extensive description of the data on the various factors potentially relevant to the diagnosis of GDM, makes it at times almost unreadable. 

In more detail: 

  • Introduction: Line 30 - 50 are not relevant at all and distract the reader.
  • Introduction should be shorter and more concise and state a clear goal of the paper so that the reader knows what lies ahead
  • Introduction: oGTT and reproducibility should be separate from the Introduction and shorter.
  • Total testing process: it would be enormously relevant and important is the authors can summarize each subsection and add their own assessment of the relative importance of the evaluated potential influence. That is not the final verdict but a helpful message to the reader.
  • It would help to describe the various factors additionally into conceptual rationale and subsequent decision on the presence of GDM for example 'screening', 'glucose load' and 'cut off points') and into 'technical issues' including stress etc.
  • The section on 'sampling' is extremely long and complicated. Why not shorten it and put the main (practical) results and consequences in a table?
  • Same for the 'Sample storage and transport'. 
  • Physiological factor: very important to have the assessment of their importance by the authors.
  • Line 316-320: we already know this; unnecessary repetition.
  • line 420-430: difficult to understand as described in this way: should be more clearly written.
  • Line 463: 'impairs glucose response'-> 'higher glucose levels'; contradictory.
  • Line 500: sleep should receive more attention. It would be easy to only do a oGTT after a good nights rest.
  • Conclusion can be much shorter as we do not have an alternative at this moment we can use.
  • In general: sometimes I feel that I read the same issue or opinion or description for the second, third or fourth time. 

Author Response

 Response to Reviewer 1

Thank you for your most valuable comments. We feel that the quality of the manuscripts has now improved considerably.

Response to comments

The current manuscript is well-researched, well-written and timely. The question is what is really want to tell the reader in current times. What's the the use of this paper?

Figure 1 is clear and basically the basis and nucleus of this paper and could be a guide to the practising clinician (or lab official) to critically assess their own way of working. That is a great benefit of this paper.

Thank you for your comment.

However, the extensive description of the data on the various factors potentially relevant to the diagnosis of GDM, makes it at times almost unreadable. 

In more detail: 

  • Introduction: Line 30 - 50 are not relevant at all and distract the reader.

Thank you for your comment. The aim of this paragraph, and the introduction itself was to offer the reader a glimpse into the history of diabetes, gestational diabetes and the OGTT helping the reader understand the evolution of diabetes and of the OGTT as a diagnostic test. This was part of the manuscript design. However, we have taken your comments on board and have reformatted and shortened the introduction.

Page 2 Lines 38-74

“Diabetes mellitus is an ancient disorder described for the first time in Egypt around 1500 BC (1). The recognition and description of diabetes began with the examination of urine. Demetrius of Apameia (100 BC) was the first to introduce the term “diabetes” which means “to pass, to go through”. Others noticed the sweet taste of urine and used this as a rudimentary test for diabetes (2). This test remained until 1776 when it was further enhanced by Dobson’s diagnostic experiments linking diabetes to a system disorder (3). The breakthrough in testing came through the work of Karl Trommer in 1841 who developed the first qualitative test for glycosuria (4) marking a shift in how diabetes was analyzed.

Diabetes with onset during pregnancy was first described by Bennewitz in 1824 in Germany (5). In 1898, Brocard showed a divergence between pregnant and non-pregnant women in relation to sugar tolerance. His research discovered that pregnant woman had less tolerance to sugar when compared to non-pregnant women (6). Lambie in 1926 determined that the first manifestation of diabetes in pregnancy occurs in the 5th or 6th month of pregnancy. He advocated the use of the 50 g oral glucose challenge test (OGCT) to calculate ketogenic-anti-ketogenic balance (7).

Based on Hoet’s study (8), in 1957, Wilkerson (9) developed a protocol proposing a 3h oral glucose tolerance test (OGTT) for patients at high risk for developing diabetes. Additionally, for women with no risk factors, he recommended a 2-step approach: 1h blood glucose measurement after a 50 g glucose load which if deemed abnormal was followed by a 3h OGTT.

The clinical equipoise regarding the best approach to screening and diagnosing GDM was the impetus for Norbet Freinkel to organise the First International Workshop on GDM in October 1979(10). A core outcome of this event was the emergence of a model for GDM screening and the suggestion that screening should be carried out between 24–28 weeks’ gestation. This model was updated in 1984, at the Second International Workshop on GDM (11) which concluded that all pregnant women should be screened for glucose intolerance with a 50g OGCT, irrespective of the time of the last meal or time of day and for diagnostic purposes the 100 g OGTT was to be employed. In November 1990, at the Third International Workshop on GDM (12) screening and diagnostic criteria were confirmed. This panel decided that the 75g 2h OGTT should be used to screen women at high risk of developing GDM (12).

The seminal Hyperglycaemia and Adverse Pregnancy Outcome (HAPO) Study in 2008 (13) addressed the importance of having all 3 glucose values (fasting, 1-h and 2-h post glucose load) in the OGTT since none of the glucose values were significantly correlated, and no single value was better in predicting the diagnosis.

The International Association of Diabetes and Pregnancy Study Groups (IADPSG) founded in 1998 to promote and enable collaboration between international groups with a focus on GDM and pre-pregnancy diabetes (T1DM and T2DM) reviewed the HAPO Study results and formulated the “Recommendations on the Diagnosis and Classification of Hyperglycaemia in Pregnancy” (14). “

  • Introduction should be shorter and more concise and state a clear goal of the paper so that the reader knows what lies ahead

Thank you for your comment. We have now shortened the introduction and a clear aim of the paper is described at the end of the introduction section.

Page 3 Lines 105-109

Despite scientists raising concerns about the reproducibility of the OGTT for over 50 years, it remains the best available test and the current ‘gold standard’ for diagnosing Type 2 Diabetes Mellitus (T2DM) and GDM. In this review, the myriad of variables that affect the reproducibility and accuracy of the OGTT will be discussed in terms of the Total Testing Process: pre-analytical phase, analytical phase and the post-analytical phase

  • Introduction: oGTT and reproducibility should be separate from the Introduction and shorter.

Thank you for your comment. For enhanced clarity, we have now formatted the introduction into 3 main parts: 1. Diabetes and gestational diabetes – historical aspects; 2. OGTT and 3. Reproducibility. We have now shortened all these sections.

Pages 2-3 Lines 38-103

“Diabetes mellitus is an ancient disorder described for the first time in Egypt around 1500 BC (1). The recognition and description of diabetes began with the examination of urine. Demetrius of Apameia (100 BC) was the first to introduce the term “diabetes” which means “to pass, to go through”. Others noticed the sweet taste of urine and used this as a rudimentary test for diabetes (2). This test remained until 1776 when it was further enhanced by Dobson’s diagnostic experiments linking diabetes to a system disorder (3). The breakthrough in testing came through the work of Karl Trommer in 1841 who developed the first qualitative test for glycosuria (4) marking a shift in how diabetes was analyzed.

Diabetes with onset during pregnancy was first described by Bennewitz in 1824 in Germany (5). In 1898, Brocard showed a divergence between pregnant and non-pregnant women in relation to sugar tolerance. His research discovered that pregnant woman had less tolerance to sugar when compared to non-pregnant women (6). Lambie in 1926 determined that the first manifestation of diabetes in pregnancy occurs in the 5th or 6th month of pregnancy. He advocated the use of the 50 g oral glucose challenge test (OGCT) to calculate ketogenic-anti-ketogenic balance (7).

Based on Hoet’s study (8), in 1957, Wilkerson (9) developed a protocol proposing a 3h oral glucose tolerance test (OGTT) for patients at high risk for developing diabetes. Additionally, for women with no risk factors, he recommended a 2-step approach: 1h blood glucose measurement after a 50 g glucose load which if deemed abnormal was followed by a 3h OGTT.

The clinical equipoise regarding the best approach to screening and diagnosing GDM was the impetus for Norbet Freinkel to organise the First International Workshop on GDM in October 1979(10). A core outcome of this event was the emergence of a model for GDM screening and the suggestion that screening should be carried out between 24–28 weeks’ gestation. This model was updated in 1984, at the Second International Workshop on GDM (11) which concluded that all pregnant women should be screened for glucose intolerance with a 50g OGCT, irrespective of the time of the last meal or time of day and for diagnostic purposes the 100 g OGTT was to be employed. In November 1990, at the Third International Workshop on GDM (12) screening and diagnostic criteria were confirmed. This panel decided that the 75g 2h OGTT should be used to screen women at high risk of developing GDM (12).

The seminal Hyperglycaemia and Adverse Pregnancy Outcome (HAPO) Study in 2008 (13) addressed the importance of having all 3 glucose values (fasting, 1-h and 2-h post glucose load) in the OGTT since none of the glucose values were significantly correlated, and no single value was better in predicting the diagnosis.

The International Association of Diabetes and Pregnancy Study Groups (IADPSG) founded in 1998 to promote and enable collaboration between international groups with a focus on GDM and pre-pregnancy diabetes (T1DM and T2DM) reviewed the HAPO Study results and formulated the “Recommendations on the Diagnosis and Classification of Hyperglycaemia in Pregnancy” (14).

  • OGTT

The OGTT has been used in clinical medicine for over 100 years (15). Jerome W. Conn was first to describe the OGTT (16). His findings were based on the work of ATB Jacobsen in 1913, who determined that carbohydrate ingestion leads to glucose fluctuations (15) but since that time, the OGTT has been contested (17). The main concerns raised by Unger in 1957 were: the diagnostic values at each time point, the timing of samples, diet (at that time 300 g of carbohydrates for 3 consecutive days prior to the test was recommended) exercise, age of the population tested, gastrointestinal factors (e.g., gastric emptying time or gastrointestinal absorption rates) and stress prior to the test that might influence the values of the test.  In 1964, Nadon et al. (18) completed a comparative analysis between OGTT and the intravenous glucose tolerance test (IVGTT) and found ‘considerable disagreement between both in the identification of diabetes. They concluded that in the future diabetes “may be diagnosed without reliance on glucose tolerance tests alone” (18).

  • Reproducibility

In 1965, McDonald et al. examined the reproducibility of the OGTT (19). In this work, 400 male volunteers free of diabetes, underwent a series of six separate OGTTs and blood glucose levels. It was determined that blood glucose levels for individuals varied considerably. A decade later these findings were corroborated by Olefsky et al. (20)

In 1991, Harlass et al. (21) found an OGTT reproducibility of only 78% in women with an elevated glucose concentration 1h post after a glucose load when repeated within 2 weeks. Catalano et al. (22) reported poor reproducibility for the OGTT in diagnosing GDM in 24% (nine of 38) of pregnant women tested. The authors hypothesized that this was likely due to a norepinephrine mediated process where maternal stress leads to increased concentrations of glucose and insulin. This theory was supported by Ko et al (23), who found the overall reproducibility of the OGTT to be 65.5% with subjects showing an improvement in glycaemic status on repeat testing. More recently, Munang et al (24) showed the reproducibility of the OGTT for GDM in a sub-Saharan African population to be 74.2%. In this study 70 women underwent the OGTT at 24-28 weeks of gestation and again one week later. However, the generalizability of the results of this study to other populations is questionable due to the small cohort size, the short time interval between repeat testing and the fact that glucose was measured on capillary blood samples and not plasma as is more usual. “

  • Total testing process: it would be enormously relevant and important is the authors can summarize each subsection and add their own assessment of the relative importance of the evaluated potential influence. That is not the final verdict but a helpful message to the reader.

Thank you for your comment

Pages 5-6 Lines 180-187

Exercise

“Despite contradictory data in the literature on the length and intensity of the exercise session, physical activity influences the way our body processes nutrients. Most of the studies on this topic have been carried out on healthy subjects or individuals with diabetes and there are no studies evaluating the impact of exercise on the antepartum OGTT results. Hence, further research is required to determine whether a single exercise session prior to the antepartum OGTT day lowers/improves glucose results. Such evidence is also necessary to ensure patient preparation for the OGTT is standardized with respect to the amount of exercise, if any, pregnant women should do in the days prior to testing.”

Page 6 Lines 204-207

Gastric emptying

“We cannot control (but should always consider) the individual variability of the rate of gastric emptying. Guidelines(1) recommend the glucose load should be drank slowly over a period of 5 minutes.  However, this is difficult to achieve and control for in clinical practice with individual wide variations in the glucose load drinking time.”

Page 7 Lines 233-236

Hydration

“Even though there are no studies examining the impact of hydration status on the OGTT in pregnancy we can extrapolate on previous findings and consider that the effects of hypo-hydration or hyper-hydration are not negligible and have the potential to lead to a misdiagnosis. Currently there are no guidelines on water intake in the days prior to the OGTT.”

Page 7 Lines 261-264

Stress

“Stress can impact glycemic status not only through hormonal responses but also through the development of unhealthy lifestyle behaviours such as overeating, smoking, increased alcohol intake (65, 66). Given the glucose response to stress and to decreased sleep duration/quality, it would seem possible that pregnant women could be erroneously diagnosed as having GDM using the OGTT.”

Page 8 Lines 290-292

Length of time spent in the fasting state

“Therefore, while there is no clear evidence regarding impact of the duration of time spent fasting  prior to the OGTT it is critical to standardize the duration spent fasting prior to laboratory OGTT and to give clear, consistent instructions to our patients to prevent inaccurate results.”

Page 8 Lines 308-311

Preparatory diet

“There is not enough evidence to recommend a pre-set diet/carbohydrate intake prior to the OGTT. Perhaps maintaining ones normal, regular diet prior to undergoing the OGTT would best reflect the individual’s capacity to metabolise glucose. However, in order to maintain a standardised approach to OGTT, possibly adherence to current guidelines is best to be recommend for now.”

Page 9 Lines 345-348

Glucose load

“Consequently, the parameters for loading dose of glucose in the tolerance test should ideally be individualized according to BSA, activity level, or necessary caloric intake calculated for the individuals basal metabolic rate in order to increase its usefulness in the identification of glucose intolerance.”

Page 10 Lines 390-393

Sampling site

“Preferably, the best model is one where blood glucose levels are reported from plasma samples where glycolysis has been delayed or inhibited. Alternatively, glucose level measurement reports should have clear information on the sample type being used and if any conversion factors had been applied in the reporting process.”

Page 11 Lines 433-436

Specimen collection tube

“In clinical practice, where delays of sample transport and processing are regularly encountered, the use of citrate tubes delivers the best option in inhibiting glycolysis and preserving the integrity of blood glucose levels ex vivo. The use of citrate buffered specimen tubes is recommended by the ADA especially if the sample processing is likely to be more than 30 minutes post-collection.”

Page 12 Lines 467-472

Sample storage and transport

“While clear recommendations exist regarding glucose sample transport and storage, the challenge is the practicality and applicability of these guidelines to the routine clinical practice settings that are not resourced for immediate sample handling and processing. Outside of research specific laboratories, worldwide, very few centers are likely equipped to adhere to such strict glucose processing methodology. Citrate buffered specimen tubes offer the best practical solution and their use is recommended by the ADA.”

Page 13 Lines 536-539

Analytical phase

“Clinicians should always seek to use accredited laboratories but must be aware that TEa does not take into account pre-analytical factors that may influence glucose results. For example, a delay of more than 4 h in processing (centrifugation and separation of plasma from blood cells) the fasting sample of the OGTT would exceed the TEa for glucose.”

Page 15 Lines 593-599

Post-analytical phase

“Inconsistencies in GDM diagnostic criteria worldwide have led to challenges in making meaningful comparisons between study results (through systematic reviews and metanalysis). Cost analysis studies should always include clinical adverse outcome prevention through diagnosis and treatment in their analysis. A very well designed study by Duran et al. (129) found that the use of the IADPSG criteria was associated with an improvement in the prevalence of maternal and neonatal adverse outcomes (pregnancy induced hypertension, prematurity, caesarean sections, NICU admissions, LGA and SGA) that was cost-effective despite a 3.5 fold rise in GDM prevalence.”

  • It would help to describe the various factors additionally into conceptual rationale and subsequent decision on the presence of GDM for example 'screening', 'glucose load' and 'cut off points') and into 'technical issues' including stress etc.

Thank you for your comment. We have now moved the screening section before the total testing process and the physiological factors prior to the pre-testing preparation factors.

Therefore, the structure of the manuscript now is

Introduction

            Diabetes and gestational diabetes – historical aspects

            OGTT

            Reproducibility

Screening

Total testing process

            Pre-analytical factors

                        Physiological factors

                                    Exercise

                                    Gastric Emptying

                                    Hydration

                                    Stress and sleep

                        Pre-testing patient preparation factors

                                    Length of time spent in fasting state

                                    Preparatory diet

                                    Glucose load

                        Pre-analysis sample handling

                                    Sampling site

                                    Specimen Collection tube

                                    Sample storage and transport

            Analytical factors

            Post-analytical Factors

Covid-19

Conclusion

  • The section on 'sampling' is extremely long and complicated. Why not shorten it and put the main (practical) results and consequences in a table? Same for the 'Sample storage and transport'. 

Thank you for your comment. We have now shortened and restructured these sections of the manuscript. And, while it is a good suggestion, we couldn’t t structure the discussion in these sections in table format.

Page 10 Lines 367-393

“Sampling site

In order to improve the interpretation of glucose results, it is imperative to understand the difference in results between samples collected from different sites (capillary plasma, capillary whole blood, venous plasma and venous whole blood). For example, the glucose levels in plasma are 11% higher than the levels in whole blood despite the fact that in clinical practice the words “plasma” and “blood” are used interchangeably (88).

Under normal physiological conditions, the post-prandial, capillary glucose levels are higher than the venous glucose levels as determined by the rate at which glucose is extracted from blood by tissues. Exploring this anomaly, Kuwa et al. (89) examined the difference in glucose levels between capillary and venous samples during the OGTT in 75 healthy individuals. They found that venous and capillary glucose levels were comparable in the fasting state, but the post-load capillary sample had significantly higher glucose levels compared to the venous one.

Stahl et al. (90) investigated whether capillary whole blood glucose levels used for analysis can be expressed as plasma results (as recommended by the ADA and WHO). Results from this study confirm that translation from capillary to plasma values may be acceptable for mean values but should not be used for individual glucose levels. These findings were confirmed by Colagiuri at al. (91) assessing the correlation between glucose levels in capillary and venous samples in fasting state, 2h after oral glucose load and random glucose levels. These authors established that both fasting and random capillary samples gave lower glucose values than venous samples but the 2h post glucose load capillary sample gave higher glucose values than the venous sample.

Adding to these research conclusions D’Orazio et al. (92), maintain that due to the difference in glucose concentrations observed between whole blood and plasma, the glucose levels are not interchangeable. They also recommend that the reporting of glucose measurements should be in plasma only as the concentration of glucose in plasma is independent of hematocrit.

Preferably, the best model is one where blood glucose levels are reported from plasma samples where glycolysis has been delayed or inhibited. Alternatively, glucose level measurement reports should have clear information on the sample type being used and if any conversion factors had been applied in the reporting process.”

Pages 11-12 Lines 438-472

“Sample storage and transport

In 1985, the WHO recommended “rapid plasma separation from samples collected in fluoride tubes” in order to prevent or delay glycolysis (103). The AACC and ADA guideline (93) recommends that samples “be immediately immersed in an ice-slurry and analyzed within 30 min of collection or rapid centrifugation after collection”. However, compliance with these guidelines is particularly challenging in the case of the OGTT due to that fact that fasting and post glucose samples are usually held at the point of patient care until the test is completed, invariably over 2h.

Consequently, diabetes prevalence will be underestimated in research studies in which sample handling and analysis is delayed as indicated by Potter et al. (104) who compared OGTT results (sodium fluoride tubes) between early centrifugation (within 10 min) and delayed centrifugation (at the end of the OGTT test) in over 12,000 women. They found the mean glucose levels for fasting, 1h, and 2h OGTT samples were higher using early centrifugation (P < 0.0001 for all) compared to delayed processing, increasing the GDM prevalence from 11.6% (n = 869/7,509) to 20.6% (n = 1,007/4,887). In the commentary accompanying this study, Price et al. (105) highlight that “without strict pre-analytical OGTT sample handling in routine clinical practice, our ability to accurately diagnose GDM and report GDM prevalence data will be flawed”.

The pre-analytical blood sampling protocol for pregnancy OGTT requires revision and standardization (106). Consideration of the difficulties that rapid centrifugation (within 30 minutes of sampling) or placement of samples on ice in busy clinics illustrates that value and pragmatism of the use of citrate blood tubes for sample collection. However, the use of citrate tubes has the potential to give a positive bias of 0.2 mmol/L, falsely increasing the rate of GDM diagnosis, such that a correction factor or revision of the diagnostic thresholds may  be required (100, 107, 108). An alternative approach could be to measure glucose in lithium heparin plasma analyzed on the critical care analyzer at the POC. In 2018, Lyons et al., recruited 12 volunteers to undergo the OGTT measuring blood glucose at each time point on the critical care analyzer (ABL90FLEX®/Glucose oxidase), and concomitantly in whole blood collected into fluoride-oxalate tubes immersed immediately in ice-slurry and analyzed within 30 minutes using the central laboratory (Roche Cobas® 8000 modular analyzer series/Hexokinase).(109) These authors demonstrated good agreement of glucose results with the WHO recommended method with results within the total allowable error analytical goal for plasma glucose of <5.5%.

While clear recommendations exist regarding glucose sample transport and storage, the challenge is the practicality and applicability of these guidelines to the routine clinical practice settings that are not resourced for immediate sample handling and processing. Outside of research specific laboratories, worldwide, very few centers are likely equipped to adhere to such strict glucose processing methodology. Citrate buffered specimen tubes offer the best practical solution and their use is recommended by the ADA.”

  • Physiological factor: very important to have the assessment of their importance by the authors.

Thank you for your comment. This has now been included in the manuscript

Page 7 Lines 261-264

“Stress can impact glycaemic status not only through hormonal responses but also through the development of unhealthy lifestyle behaviours such as overeating, smoking, increased alcohol intake (65, 66). Given the glucose response to stress and to decreased sleep duration/quality, it would seem possible that pregnant women could be erroneously diagnosed as having GDM using the OGTT.”

  • Line 316-320: we already know this; unnecessary repetition.

Thank you for your suggestion. This paragraph has now been removed from the manuscript.

  • line 420-430: difficult to understand as described in this way: should be more clearly written.

Thank you for your comment. We have now re-written the paragraph

Page 5 Lines 164-171

“Slentz et al. (38) studied the effects of different intensities of exercise on the OGTT in individuals with prediabetes. These authors found significant reductions in fasting glucose levels only when low amount of moderate exercise and diet was combined. Higher levels of exercise were associated with improved glucose concentrations at 30 minutes post OGTT but was less effective when compared to the combination of diet and exercise. When overall improvement in glucose tolerance was assessed, low amounts of moderate exercise alone was determined to be half as effective as diet and exercise combined but twice as effective as high amounts of high intensity exercise.”

  • Line 463: 'impairs glucose response'-> 'higher glucose levels'; contradictory.

Thank you for this excellent observation. This has now been rectified in the text

Page 6 Lines 213-215

Johnson et al. (99) had similar findings concluding that 3 days of low total water intake in people with T2DM acutely modifies blood glucose response during an OGTT with higher glucose levels in the hypohydrated group.”

  • Line 500: sleep should receive more attention. It would be easy to only do a oGTT after a good nights rest.

Thank you for your comment. We have now expanded our section on sleep.

Page 7 Lines 254-260

“The association between sleep duration and quality and glucose homeostasis has been highlighted by additional studies (61, 62) which found that shorter sleep duration is associated with higher glucose levels, particularly the fasting and 2h glucose level on the OGTT. Retrukatul et al. (63) found that pregnant women with reduced sleep duration (less than 7h per night) have an increased risk of developing GDM; in fact, each hour of reduced sleep leads to a 4% increase in blood glucose levels. These results are supported by Myoga et al (64) who also found that pregnant women that sleep less than 5h per night had higher random blood glucose levels.”

  • Conclusion can be much shorter as we do not have an alternative at this moment we can use.

Thank you for your comment. We have now moved the COVID 19 paragraph into a separate section. The conclusion now reiterates the need of improving the way we perform the OGTT and the dire need for a new diagnostic test while enumerating a few of the emerging biomarkers that have this potential.

Pages 15-16 Lines 625-634

“The OGTT is subject to several factors spanning the total testing process that have the potential to influence its results and negatively impact patient care. Clear guidance is needed to ensure a universal standardized approach to performing and interpreting the OGTT for the diagnosis of GDM. This will permit global harmonization of the detection of GDM, improve the accuracy and reproducibility of the OGTT and provide for better outcomes for mothers and their offspring. Alongside this, the search for better biomarkers to diagnose GDM is gaining pace with several biomarkers currently under evaluation: adiponectin (136), sex hormone binding globulin (137), plasma glycated CD59 (138), 1,5 anhydroglucitol (139), plasminogen activator inhibitor-1 (140) and visfatin (141) to name but a few. The diagnostic accuracy and clinical usefulness of many of these novel biomarkers remain to be fully validated. “

  • In general: sometimes I feel that I read the same issue or opinion or description for the second, third or fourth time. 

We hope that the revision and reformatting of the manuscript following your most valuable suggestions have improved this.

Reviewer 2 Report

In this article, the AAs extensively examine the history of diagnosis of T2DM with OGTT and have a specific focus on the need of a correct diagnosis of GDM.

The paper is interesting and the work exhaustive.

The last section evaluates the diagnosis of GDM during the COVID19 pandemic. I suggest that you integrate the chapter with the following references

1: Thangaratinam S, Cooray SD, Sukumar N, Huda MSB, Devlieger R, Benhalima K, McAuliffe F, Saravanan P, Teede HJ. ENDOCRINOLOGY IN THE TIME OF COVID-19: Diagnosis and management of gestational diabetes mellitus. Eur J Endocrinol. 2020 Aug;183(2):G49-G56. doi: 10.1530/EJE-20-0401. PMID: 32454456.

2: van Gemert TE, Moses RG, Pape AV, Morris GJ. Gestational diabetes mellitus testing in the COVID-19 pandemic: The problems with simplifying the diagnostic process. Aust N Z J Obstet Gynaecol. 2020 Jul 13:10.1111/ajo.13203. doi: 10.1111/ajo.13203. Epub ahead of print. PMID: 32662072; PMCID: PMC7405039.

3: Torlone E, Festa C, Formoso G, Scavini M, Sculli MA, Succurro E, Sciacca L, Di Bartolo P, Purrello F, Lapolla A. Italian recommendations for the diagnosis of gestational diabetes during COVID-19 pandemic: Position statement of the Italian Association of Clinical Diabetologists (AMD) and the Italian Diabetes Society (SID), diabetes, and pregnancy study group. Nutr Metab Cardiovasc Dis. 2020 Aug 28;30(9):1418-1422. doi: 10.1016/j.numecd.2020.05.023. Epub 2020 May 29. PMID: 32675009; PMCID: PMC7258852.

4: van-de-l'Isle Y, Steer PJ, Watt Coote I, Cauldwell M. Impact of changes to national UK Guidance on testing for gestational diabetes screening during a pandemic: a single centre observational study. BJOG. 2020 Sep 5. doi: 10.1111/1471-0528.16482. Epub ahead of print. PMID: 32888369.

Author Response

Response to Reviewer 2

Thank you for your most valuable comments. We feel that the quality of the manuscripts has improved considerably.

Response to comments

In this article, the AAs extensively examine the history of diagnosis of T2DM with OGTT and have a specific focus on the need of a correct diagnosis of GDM.

The paper is interesting and the work exhaustive.

Thank you very much for your comment.

The last section evaluates the diagnosis of GDM during the COVID19 pandemic. I suggest that you integrate the chapter with the following references

1: Thangaratinam S, Cooray SD, Sukumar N, Huda MSB, Devlieger R, Benhalima K, McAuliffe F, Saravanan P, Teede HJ. ENDOCRINOLOGY IN THE TIME OF COVID-19: Diagnosis and management of gestational diabetes mellitus. Eur J Endocrinol. 2020 Aug;183(2):G49-G56. doi: 10.1530/EJE-20-0401. PMID: 32454456.

2: van Gemert TE, Moses RG, Pape AV, Morris GJ. Gestational diabetes mellitus testing in the COVID-19 pandemic: The problems with simplifying the diagnostic process. Aust N Z J Obstet Gynaecol. 2020 Jul 13:10.1111/ajo.13203. doi: 10.1111/ajo.13203. Epub ahead of print. PMID: 32662072; PMCID: PMC7405039.

3: Torlone E, Festa C, Formoso G, Scavini M, Sculli MA, Succurro E, Sciacca L, Di Bartolo P, Purrello F, Lapolla A. Italian recommendations for the diagnosis of gestational diabetes during COVID-19 pandemic: Position statement of the Italian Association of Clinical Diabetologists (AMD) and the Italian Diabetes Society (SID), diabetes, and pregnancy study group. Nutr Metab Cardiovasc Dis. 2020 Aug 28;30(9):1418-1422. doi: 10.1016/j.numecd.2020.05.023. Epub 2020 May 29. PMID: 32675009; PMCID: PMC7258852.

4: van-de-l'Isle Y, Steer PJ, Watt Coote I, Cauldwell M. Impact of changes to national UK Guidance on testing for gestational diabetes screening during a pandemic: a single centre observational study. BJOG. 2020 Sep 5. doi: 10.1111/1471-0528.16482. Epub ahead of print. PMID: 32888369.

Thank you for your suggestion. We have added the recommended references and expanded our discussion on the topic.

Page 15 Lines 610-615

Gemert et al (133) have shown that by  only using a fasting plasma glucose ≤ 4.6 mmol/l for the diagnosis of GDM, 29% of women would have been missed. Similarly, van-de- l’Isle et al. (134) found that by using the Royal College of Obstetrics and Gynecologists recommendations for the diagnosis of GDM ( fasting glucose ≥ 5.3 mmol/l or HbA1c ≥ 39 mmol/mol or random plasma glucose ≥ 9 mmol/l), 57% of women would have been wrongly diagnosed as not having GDM. “